# RSAD2 Is an Effective Target for High-Yield Vaccine Production in MDCK Cells

**DOI:** 10.3390/v14112587

**Published:** 2022-11-21

**Authors:** Zilin Qiao, Yuejiao Liao, Mengyuan Pei, Zhenyu Qiu, Zhenbin Liu, Dongwu Jin, Jiayou Zhang, Zhongren Ma, Xiaoming Yang

**Affiliations:** 1Gansu Tech Innovation Center of Animal Cell, Biomedical Research Center, Northwest Minzu University, Lanzhou 730030, China; 2Gansu Provincial Bioengineering Materials Engineering Research Center, Lanzhou Minhai Bio-Engineering Co., Ltd., Lanzhou 730030, China; 3Life Science and Engineering College, Northwest Minzu University, Lanzhou 730030, China; 4National Engineering Technology Research Center for Combined Vaccines, Wuhan 430207, China; 5Wuhan Institute of Biological Products Co., Ltd., Wuhan 430207, China; 6China National Biotech Group Company Limited, Beijing 100029, China

**Keywords:** DIA, MDCK, influenza virus, RSAD2, vaccine

## Abstract

Increasingly, attention has focused on improving vaccine production in cells using gene editing technology to specifically modify key virus regulation-related genes to promote virus replication. In this study, we used DIA proteomics analysis technology to compare protein expression differences between two groups of MDCK cells: uninfected and influenza A virus (IAV) H1N1-infected cells 16 h post infection (MOI = 0.01). Initially, 266 differentially expressed proteins were detected after infection, 157 of which were upregulated and 109 were downregulated. We screened these proteins to 23 genes related to antiviral innate immunity regulation based on functional annotation database analysis and verified the mRNA expression of these genes using qPCR. Combining our results with published literature, we focused on the proteins RSAD2, KCNN4, IDO1, and ISG20; we verified their expression using western blot, which was consistent with our proteomics results. Finally, we knocked down RSAD2 using lentiviral shRNA expression vectors and found that RSAD2 inhibition significantly increased IAV NP gene expression, effectively promoting influenza virus replication with no significant effect on cell proliferation. These results indicate that RSAD2 is potentially an effective target for establishing high-yield vaccine MDCK cell lines and will help to fully understand the interaction mechanism between host cells and influenza viruses.

## 1. Introduction

Influenza, which is one of the most common respiratory diseases, is caused by influenza virus infection. Outbreaks tend to be seasonal, typically occurring in winter, spring, and autumn, and only occasionally in summer; furthermore, these outbreaks can cause a considerable public health burden [1]. Currently, influenza vaccination is an effective means to prevent seasonal influenza virus infection, which can significantly reduce the risk of disease and serious complications. Compared with the traditional chicken embryo matrix, the cell matrix influenza vaccine can significantly improve vaccine production efficiency and provide more protection [2]. Additionally, the cell matrix influenza vaccine eliminates the protein allergy risks and fertilized egg supply restrictions associated with traditional chicken embryo matrix vaccines [3,4]. Furthermore, the development of domestication of suspension cells and serum-free suspension culture technology has provided a flexible and controlled cell matrix influenza vaccine production process [5,6,7].

MDCK cells are highly sensitive to virus infection and resistant to mutation. They can support large-scale cell culture and high-yield influenza virus production, and have been used for influenza vaccine production in several countries [8,9]. Recent developments in gene editing technology have led to increased research in generating cell lines through genetic modifications that will be more suitable for vaccine production [10,11]. In MDCK cells, overexpression of the DRI gene reduces the activation of the JAK-STAT pathway, which inhibits interferon-stimulated gene (ISG) expression, and subsequently promotes influenza virus replication [12]. In ISG knockout Vero cells, the total particle yield of influenza A virus (IAV) was 70 times higher than wild-type Vero cells, and the ATCC-modified STAT1 knockout cell line increased the viral titer of IAV by thirty times. To obtain more effective genetic modification targets for improved vaccine production, we need a more comprehensive understanding of the molecular mechanisms involved in the virus–host interaction during infection.

Throughout the entire process of influenza virus infection—from viral invasion of host cells to complete virus adsorption, integration, and release—numerous host cell immune response-related genes alter their expression and affect virus replication by regulating antiviral and other signaling pathways [13]. Most of these genes play antiviral roles by inhibiting viral infection and proliferation; however, some host genes provide favorable conditions for virus proliferation by inhibiting the expression of host antiviral factors. The innate immune response is an important line of defense in host cells against viral infection [14,15]. Host cell innate immune response pathways against influenza virus commonly include RIG-I-like receptors (RLRs), Toll-like receptors (TLRs), JAK-STAT, NF-kB, and other pattern recognition receptor (PRR)-mediated signaling pathways [16], which have the potential to serve as genetic engineering targets to promote virus proliferation and vaccine production. For example, the stable knockout of IRF7, which is a component of the TLR signaling pathway, can produce a high titer of influenza virus in MDCK cells [17]; additionally, siat7e gene overexpression in adherent or suspended MDCK cells results in increased hemagglutinin (HA) antigen production [18]. RSAD2 is an interferon-inducible protein that exhibits broad-spectrum antiviral activity. For example, RSAD2 can inhibit hepatitis C virus by interacting with viral nonstructural protein 5A (NS5A) [19]; affect the production and accumulation of dengue virus type 2 RNA by interacting with the viral protein NS3; inhibit the proliferation of the rabies virus by reducing the production of cholesterol and sphingomyelin; and inhibit the cellular release of the influenza virus capsid by interfering with lipid raft metabolism [20]. However, further study is required to determine whether RSAD2 can be a potential genetic engineering target to improve influenza virus production.

To comprehensively identify the critical genes involved in influenza virus proliferation in MDCK cells, and to screen these target genes for their effectiveness in constructing genetically engineered high-yield vaccine production cell lines, we used data independent acquisition (DIA) proteomic analysis technology in this study to detect protein expression changes in MDCK cells before and after infection with IAV. Preliminary screening at the mRNA and protein levels identified four virus response-related proteins, which were verified with real-time quantitative PCR (qPCR) and western blotting. Furthermore, these genes were knocked down using RNAi technology and the resulting changes in virus replication level and cell proliferation were measured in order to evaluate the potential for each gene to serve as an effective gene editing target to improve vaccine production. These results will improve understanding of the interaction mechanism between host cells and influenza viruses, and provide a basis for screening potential gene editing targets for establishing a new high-yield influenza vaccine MDCK cell line.

## 2. Materials and Methods

### 2.1. Cells and Virus

MDCK adherent cells (#CCL-34, ATCC, Manassas, VA, USA) were provided by the Animal Cell Engineering Center of Gansu Province, China. The cells were maintained in DMEM containing 10% NBS and cultured at 37 °C and 5% CO_2_. Influenza A/Puerto Rico/8/34 (A/PR/8/34) H1N1 virus, A/Texas/50/2012(H3N2)NYMCX-223A, B/Colorado/06/2017-like virus B (Victoria lineage), and B/Phuket/3073/2013-like virus B (Yamagata/16/88 lineage) were obtained from Wuhan Institute of Biological Products Co., Ltd. (Wuhan, China).

### 2.2. Protein Extraction and Quantification

MDCK cells were cultured in a T75 cell culture flask to 90% confluence and H1N1 virus diluent was added (MOI = 0.01, 2 μg/mL TPCK trypsin). After 16 h, the cells were harvested by scraping, and 1 mL of a 1× solution containing an appropriate amount of SDSL3 and EDTA was added and placed on ice for 5 min. After adding 500 µL 10 mM DTT, the cells were subjected to ultrasonic crushing (60 Hz) for 2 min, followed by centrifugation at 25,000× *g* at 4 °C for 15 min. An additional 500 µL 10 mM DTT was added to the resulting supernatant, and placed in a water bath at 56 °C for 1 h. IAM was then added to a final concentration of 55 mM and stored for 45 min in the dark. Cold acetone was then added to the protein solution at a ratio of 1:5 and stored at −20 °C for 30 min followed by centrifugation at 25,000× *g* at 4 °C for 15 min. The supernatant was discarded; after air-drying the precipitate, the proper amount of SDSL3 was added, and ultrasonic crushing (60 Hz, 2 min) was performed followed by centrifugation at 25,000× *g* at 4 °C for 15 min. Finally, the protein concentration was measured from the resulting supernatant using the Bradford protein assay.

### 2.3. Protein Digestion and Peptide Desalting

To digest the extracted proteins, 2.5 μg trypsin was added to each 100 μg protein sample to achieve a 40:1 ratio of protein:enzyme, and the solution was maintained at 37 °C for 4 h for enzymatic hydrolysis. The enzymatically hydrolyzed peptides were desalted on the Strata-X column (Phenomenex, Torrance, CA, USA) and vacuum dried. The peptide fragments were separated and dried using the LC-20AD HPLC system (Shimadzu, Kyoto, Japan), and data-dependent acquisition (DDA) library building and DIA quantitative detection analyses were performed using Nano-LC-MS/MS (Q Exactive HF-X Mass Spectrometer (ThermoFisher Scientific, Waltham, MA, USA)). Mass spectrometry and protein identification services were provided by BGI (China). The mass spectroscopic proteomic data are shown in Appendix A.

### 2.4. qPCR Analysis

Total RNA was extracted with TRIzol reagent (Shanghai Yamei Biomedical Technology Co., Ltd.) and reverse transcribed to generate cDNA according to the manufacturer’s instructions (China Hunan Accurate Bio-Medical Technology Co., Ltd.). To detect changes in gene expression, the qPCR mixture was prepared with a 50 ng cDNA template, primers, and SYBR Green fluorescent quantitative PCR (Wuhan ABclonal Biotechnology Co., Ltd.). The reaction was performed using the following conditions: 95 °C initial denaturation for 15 min; 40 cycles of denaturation at 95 °C for 10 s, annealing at 60 °C for 20 s, and extension at 72 °C for 30 s; and melting curve analysis after amplification. GAPDH was used as the internal reference gene, and the relative gene expression was calculated using the 2^-ΔΔCt^ method. The primer and probe sequences are shown in Appendix A.

### 2.5. Western Blot Analysis

Each cell sample was combined with a 500 μL PMSF/RIPA buffer mixture (PMSF: RIPA = 1:100) and proteins were quantified using the BCA protein assay. To denature the proteins, 75 µL of each protein sample were added to 20 µL 5× loading buffer and placed in a water bath at 100 °C for 10 min. The denatured proteins were separated using SDS-PAGE (120 V, 120 min) with 7.5–15% precast gels (Bio-Rad, Hercules, CA, USA) and transferred to membranes (220 mA, 90 min). The membranes were blocked with 5% skim milk powder and sealed at room temperature for 2 h, incubated with the primary antibody (Proteintech, 1:1000 dilution) at room temperature for 2 h, washed with TBST, incubated with secondary HRP-labeled goat anti-mouse IgG or goat anti-rabbit IgG antibody (1:5000 dilution) at room temperature for 1 h, washed with TBST, and visualized after ECL color development (Sinsitech, MiniChemi 610, China) to determine relative protein expression. RIPA lysis buffer (PC101), PMSF (GRF101), and a BCA Protein Quantitative Kit (ZJ101) were purchased from Shanghai Yamei Biomedical Technology Co., Ltd.; SDS-PAGE loading buffer, 5× (P1040), and a WB gel making kit (A1010) were purchased from Beijing Solarbio Science & Technology Co., Ltd.; Anti RSAD2 rabbit monoclonal antibody (Cat No. 28089-1-AP), Anti KCNN4 rabbit monoclonal antibody (Cat No. 23271-1-AP), Anti IDO1 rabbit monoclonal antibody (Cat No. 13268-1-AP), Anti ISG20 rabbit monoclonal antibody (Cat No. 22097-1-AP), Anti GAPDH mouse monoclonal antibody (Cat No: 60004-1-Ig), Goat anti-mouse IgG (cat: SA00001-1), and Goat anti-rabbit IgG (cat: SA00001-2) were purchased from Proteintech Group, Inc; Anti IBV NP rabbit monoclonal antibody (cat: B017) was purchased from Abcam.

### 2.6. RNA Interference

The lentiviral vector was provided by Shanghai Jikai Biotechnology Co., Ltd. Wild-type MDCK cells were inoculated into a 12-well plate at 1 × 10^5^ cells/well. The lentivirus solution was added (MOI = 100) when the cells reached 50–70% confluence. After 12 h, the medium was changed to DMEM supplemented with 10% NBS and incubated in 5% CO_2_ at 37 °C for 48 h. The fluorescence expression was then observed under a fluorescence microscope and the cells were subcultured. After cell adherence, DMEM supplemented with 4% NBS was added and 4 μg/mL puromycin was used to screen the cell resistance.

### 2.7. Influenza Virus Infection

USP1-knockdown or -overexpression cells were inoculated into a 6-well plate at 6 × 10^5^ cells/well and incubated at 37 °C and 5% CO_2_ for 24 h. The culture medium was discarded, the cells were washed three times with PBS, and 1 mL serum-free DMEM culture solution with diluted Influenza A/Puerto Rico/8/34 (A/PR/8/34) H1N1 virus, A/Texas/50/2012(H3N2) NYMCX-223A, B/Colorado/06/2017-like virus B (Victoria lineage), and B/Phuket/3073/2013-like virus B (Yamagata/16/88 lineage) (2 μg/mL TPCK trypsin, MOI = 0.01) was added. After infection for 2 h, the virus supernatant was discarded, additional serum-free DMEM (2 μg/mL TPCK trypsin) was added, and the virus supernatant and cell samples were collected at 12, 24, 36, 48, and 60 h post infection (hpi) for further study.

### 2.8. SH-RSAD2 Cell Growth Curve

Sh-RSAD2, sh-control, and wild-type MDCK cells were used to make the cell suspension. The cells were diluted using DMEM supplemented with 10% NBS to 5000 cells/mL and inoculated into 24-well plates at a volume of 1 mL/well, each group containing three biological replicates tested in parallel. The cells were cultured at 5% CO_2_ and 37 °C; and three wells were counted at the same time every day over 8 days. After trypsin digestion, twenty microliters of this cell suspension were then counted using a hemocytometer (Count star Biotech, #IC1000). For each well, three counts were performed, and an average was taken.

### 2.9. TCID_50_

After adding 1 × 10^4^ MDCK cells to a 96-well plate, the plate was cultured at 37°C and 5% CO^2^ for 24 h. We discarded the culture medium, washed the cells twice with PBS, obtained the viral supernatant to be tested in the 6-well plate, and used the virus maintenance solution (1 μg/mL trypsin DMEM) for logarithmic dilution. The dilution degree was 10**^−^**^1^~10**^−^**^11^ in turn; different dilutions of virus solution were added to columns 1–11 (100 μL/well) of the 96-well cell plate (each group of virus has three independent biological duplicate samples (*n* = 3 per group)); PBS (100 μL/well) was added to column 12 as the negative control. The cells were incubated in a 5% CO^2^ incubator at 34 °C for 72 h. An amount of 100 μL crystal violet staining was added for 10 ~ 20 min. The results were judged according to the cytopathic effect (CPE). If >50% of the area at the bottom of the hole was stained purple, the hole was considered negative.

### 2.10. Statistical Analysis

Data were analyzed with GraphPad Prism 5.0 using one-way ANOVA or Student’s *t*-test and presented as the mean ± standard deviation (SD) of three independent experiments. A *p*-value < 0.05 was considered statistically significant with additional significance threshold values defined as follows: * *p* < 0.05; ** *p* < 0.01; *** *p* < 0.001.

## 3. Results

### 3.1. Differentially Expressed Proteins Associated with H1N1 Influenza Virus Infection

We used the Q Exactive HF-X Mass Spectrometer (ThermoFisher Scientific, Waltham, MA, USA) in the DIA mode to analyze the differential protein expression between two groups of MDCK cells: uninfected (MDCK mock) or infected with Influenza A/PR/8/34 H1N1 for 16 h (MDCK H1N1 16 hpi). Each group of cells has three independent biological duplicate samples (*n* = 3 per group); R software with the MSstats package was used to quantify the differentially expressed peptides and proteins. Our initial analysis identified 5817 differentially expressed proteins. We screened the differentially expressed proteins between the two groups based on two conditions: logFC > 1 and *p*-value < 0.05, as determined using the student’s *t*-test. We found 266 differentially expressed proteins between uninfected MDCK mock cells and IAV-infected MDCK H1N1 16 hpi cells, as shown in the volcano plot (Figure 1A). Among these proteins, 157 were upregulated and 109 were downregulated. Furthermore, our differential gene cluster heatmap shows good consistency in protein expression patterns among biological replicates within the same group, and strong repeatability with significant protein expression differences between the two groups (Figure 1B). These results indicate that the differential expression of these proteins strongly correlates with viral infection-associated biological processes.

### 3.2. Cluster Analysis of Differentially Expressed Proteins

Gene ontology (GO) is an international standardized gene function classification system, which provides a set of dynamically updated standard vocabularies to comprehensively describe the attributes of genes and gene products in organisms. The GO system comprises three ontologies, namely, the molecular function, cellular component, and biological process of genes. We assigned 9445 GO enrichment classification items to our 266 differentially expressed proteins and ranked the proteins with significant differences among groups according to the number of clustering proteins. Within the biological process category, most of the differentially expressed proteins are involved in the regulation of cell metabolism, signal transduction, and cell proliferation (Figure 2A). Notably, numerous differentially expressed proteins are involved in the cellular immune response to exogenous stimuli, which will be our focus. The cell location clustering results showed that most of the differentially expressed proteins were distributed in various organelles and membrane structures, along with extracellular and intercellular junction structures. Within the molecular function category, many differential proteins are related to binding activity and catalytic activity. According to the KEGG database, differential genes are classified according to the different signaling pathways in which they participate. In addition to the pathways directly related to immune response, the cluster analysis results also identified links with cell killing and detoxification pathways, suggesting the importance of other biological processes that may be significantly affected by virus infection or that may participate in virus regulation.

### 3.3. Screening of Immune Response-Related Proteins

According to the functional annotation of proteins in our GO enrichment analysis and KEGG and Uniprot databases, we screened 42 differential proteins related to cellular innate immunity and viral response and interaction, including 28 upregulated proteins and 14 downregulated proteins. Considering the possibility of previously unreported proteins involved in virus regulation in MDCK cells, we also included the 10 most upregulated and 10 most downregulated proteins in our analysis, based on a *p*-value < 0.01. Our clustering analysis results revealed that the screened differential proteins related to the immune response are primarily involved in infectious diseases processes, such as viral infection, signal transformation, bacterial infection, and immune system diseases (Figure 2B). The KEGG analysis results further reveal that these proteins are primarily involved in human cytogenetic viruses in the section, Epstein–Barr virus infection, the NF-κB signaling pathway, antigen processing and presentation, and viral myocarditis, and that they play an important role in the viral immune response (Figure 2C).

### 3.4. Validation of mRNA Expression

Since high-throughput mass spectrometry analysis can yield false positive results, we used qPCR to verify the expression of our 23 screened proteins at the mRNA level (see Table 1). Our results showed that the gene expression of proteins was consistent with our proteomics analysis (Figure 3). Furthermore, 14 proteins that were upregulated after virus infection—including TNFaIP3, HNRNPH2, RSAD2, ISG20, IDO1, and KCNN4—also exhibited significantly increased mRNA levels relative to uninfected cells (*p* < 0.05). These results suggest that these proteins are likely to participate in the important stage of MDCK cells regulating virus replication and play a key regulatory role.

### 3.5. Validation of Protein Expression

The host immune response activated by influenza virus infection is critical for inhibiting the virus infection. In particular, the host-cell-mediated innate immune response is a vital anti-virus defense system [21]. In this pathway, interferon binds with its receptor, which activates the downstream JAK-STAT signaling pathway, ultimately inducing interferon stimulating factors that target different steps of the virus growth cycle [22]. RSAD2, IDO1, and ISG20 are typical interferon stimulating factors that can regulate viral mRNA expression and protein translation [23,24,25]. We performed western blots to verify the protein level of RSAD2, IDO1, ISG20, and KCNN4. Consistent with our proteomics and mRNA analysis results, our western blot results showed that RSAD2, IDO1, ISG20, and KCNN4 protein levels in IAV-infected MDCK cells were also significantly higher than uninfected MDCK cells (*p* < 0.05; Figure 4).

### 3.6. RSAD2 Knockdown Increased Influenza Virus Titers in MDCK Cells

Collectively, our data indicated that IAV H1N1 infection induces RSAD2 expression in MDCK cells. The expression of RSAD2 increased with the prolongation of virus infection time, and reached the maximum at 36–48 hpi. (Figure 5A–C). Therefore, in order to explore whether RSAD2 impacts influenza virus replication in MDCK cells, we designed an shRNA targeting RSAD2 and established sh-RSAD2 cells. To verify that the knockdown was successful, we confirmed that the mRNA and protein levels in sh-RSAD2 cells were significantly reduced relative to controls (Figure 5D,F). Next, we infected sh-RSAD2 and sh-control cells with Influenza A/Puerto Rico/8/34 (A/PR/8/34) H1N1 virus, A/Texas/50/2012 (H3N2) NYMCX-223A, B/Colorado/06/2017-like virus B (Victoria lineage), and B/Phuket/3073/2013-like virus B (Yamagata/16/88 lineage) (MOI = 0.01); detected the proliferation of influenza virus in the cells; collected samples at 12, 24, 36, 48, and 60 hpi; and detected the viral NP protein using qPCR and western blot. Compared with the sh-control cells, RSAD2 knockdown promoted both the mRNA and protein expression of the influenza virus NP gene during infection (Figure 5G,I). TCID_50_ results showed that the knockdown of RSAD2 could significantly promote the replication and proliferation of Influenza A/Puerto Rico/8/34 (A/PR/8/34) H1N1 virus and B/Phuket/3073/2013-like virus B (Yamagata/16/88 lineage) in MDCK cells. The H1N1 virus titer reached the maximum at 48 hpi, and the BY virus titer reached the maximum at 36 hpi (Figure 5J,K). In order to further clarify how RSAD2 mediates the replication and proliferation of the influenza virus in cells, we preliminarily detected the gene expression of downstream factors mediated by RLR and TLR receptors in sh-RSAD2 cells using qPCR. We found that the expression of the common antiviral factor RIG-1, IFN-α, IL6, MXA, and IL1 in sh-RSAD2 cells was significantly inhibited after virus infection compared with sh-control cells (Figure 5L). These results indicate the potential feasibility of establishing genetically engineered vector cells conducive to virus proliferation by inhibiting the expression of key genes in the host cell immune response signaling pathway, thereby increasing the vaccine production capacity of cell lines.

## 4. Discussion

Influenza pandemics pose a serious threat not only to human life and health but also to social and economic development. For instance, acute respiratory distress syndrome (ARDS) caused by influenza virus infection is an important cause of human and animal death, and severe pulmonary edema is particularly prominent in patients with acute and complex influenza infection [26]. A timely provision of an adequate and effective influenza vaccine is the preferred method for countries to mitigate the threats associated with influenza pandemics [27,28,29]. Influenza vaccines currently on the market include three types: inactivated influenza vaccine, live attenuated influenza vaccine, and recombinant protein vaccine. Among these three types, the inactivated influenza vaccine is the most widely used due to its high degree of safety and mature production technology [30]. However, the current influenza vaccine still has many limitations, including long production cycles, risk of allergic reactions, and limited protection scope since the vaccine is not broad-spectrum [31]. The effectiveness of the vaccine relies on its ability to provide protection against circulating virus strains that match the vaccine; however, due to antigen drift or antigen conversion of circulating influenza virus strains, the vaccine strains often do not match the epidemic strains, thereby limiting the protection provided by the current influenza vaccine [32]. Therefore, a new generation of a universal influenza vaccine would have many benefits over the current influenza vaccine, namely, it could provide more extensive protection from most influenza viruses, effectively induce humoral and cellular immunity against conservative epitopes of influenza viruses, provide broad-spectrum protection from various types or subtypes of influenza viruses, be safe and effective, and have a rapid production platform [33,34,35].

The host interferon antiviral response is a crucial line of defense against virus infection. After virus infection, host PRRs recognize virus products and induce the expression of various antiviral ISGs to inhibit virus replication [36]. RSAD2 is a broad-spectrum anti-virus ISG protein, which can prevent virus replication by affecting the expression of specific enzymes and has been shown to inhibit the expression of many membranes’ fusion viruses, including influenza virus, respiratory syncytial virus, hepatitis C virus, and HIV-1 [19,20]. Additionally, KCNN4, a member of the KCNN family, has been shown to have increased expression in response to IAV H1N1 influenza virus infection, which in turn leads to increased expression of anti-inflammatory factors and antiviral ISGs in epithelial cells [37]. In our study, influenza virus infection significantly increased both RSAD2 and KCNN4 protein expression; furthermore, RSAD2 knockdown significantly increased expression of the IAV H1N1 NP gene at both the protein and mRNA level in MDCK cells. Collectively, these results suggest that suppressing the expression of key immune response-related genes can lead to the establishment of improved influenza vaccine-producing MDCK cells.

The host innate immune response is an ideal target for the development of a new generation of MDCK-producing vaccines. The universal influenza vaccine has previously been designed based primarily on the viral genes HA, NA, and M2, as well as the T cell immune response induced by the viral matrix proteins NP, PA, PB1, PB2, and M1 [38,39]. By using DIA proteomics technology to search for proteins related to influenza virus infection, and by exploring the components of the host’s natural immune response system mediated by these proteins, future studies can target the signaling pathways mediated by multiple PRRs, such as RLRs, TLRs, and NOD-like receptors. This research may not only improve the vaccine production capacity of MDCK and other vaccine production cells, but may also provide new antiviral targets for the treatment of viral infection.

## 5. Conclusions

In this study, we proved that the comprehensive proteomic analysis of MDCK cells before and after H1N1 influenza virus infection using the DIA method is a useful tool to identify critical host cell proteins with regulatory effects on H1N1 infection. The data revealed the activation of several biological processes, including components of the IAV H1N1 infection-induced innate immune response: specifically, the expression of a variety of antiviral ISGs, the activation of the autophagic pathway, and the production of inflammatory factors. In addition to several proteins previously shown to be regulated in H1N1 infection, we also identified previously unreported IAV-induced proteins, such as PMSD10, POLR3F, and DCTN3; it will require further research to uncover their mechanisms. We propose that the proteomic profile provided in this study can be used to uncover potential vaccine targets and provide ideas for the establishment of new MDCK cell lines for high-yield influenza vaccine production using genetic engineering methods.

## Figures and Tables

**Figure 1 viruses-14-02587-f001:**
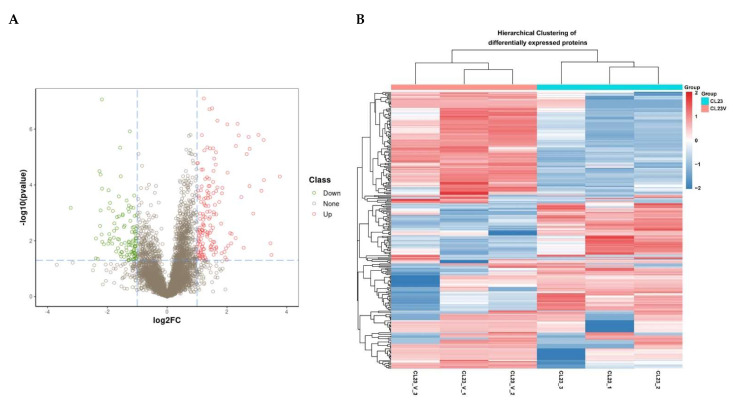
Proteomic - analysis of MDCK cells infected with Influenza A virus (IAV) H1N1 relative to uninfected MDCK cells. (**A**) The volcano plot represents the significance and magnitude of protein level changes in uninfected MDCK cells (MDCK mock) and IAV H1N1-infected MDCK cells 16 h post infection (hpi) (MDCK H1N1 16 hpi). Green dots represent downregulated differentially expressed proteins, red dots represent upregulated differentially expressed proteins, and “None” represents proteins with no significant expression differences between the two groups of cells. (**B**) The heat map represents significant protein differences between MDCK mock and MDCK H1N1 16 hpi cells. The color scale indicates the relative protein abundance, with darker shades representing the greatest difference in protein abundance between the two groups; red indicates up-regulation and blue indicates downregulation. The student’s *t*-test was used to identify statistical significance.

**Figure 2 viruses-14-02587-f002:**
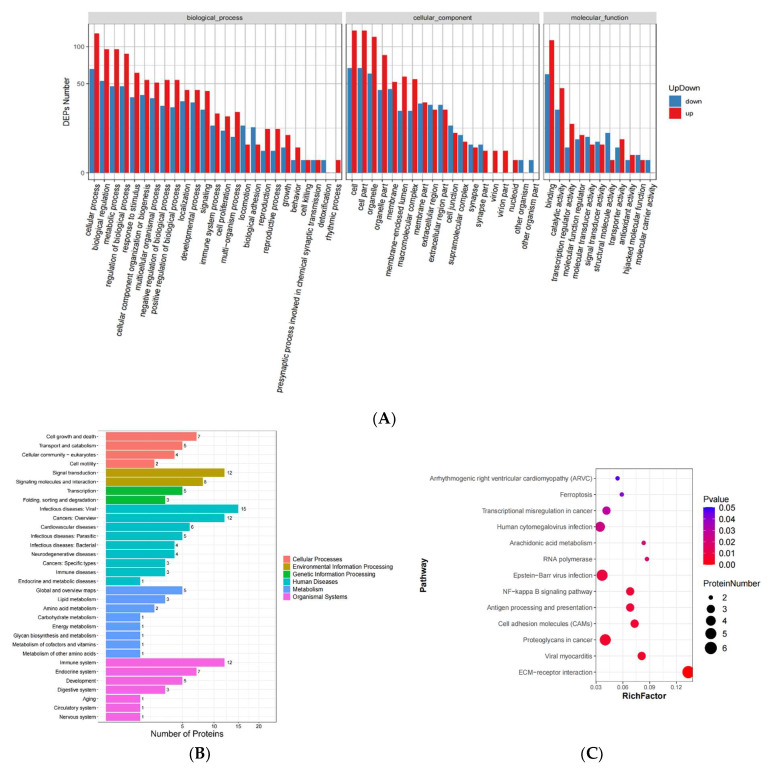
Screening of proteins related to the immune response of MDCK cells after IAV H1N1 infection. (**A**) Preliminary GO results for all 266 differentially expressed proteins, which are classified according to biological process, cellular component, and molecular function. (**B**) The GO results of 42 differentially expressed proteins after screening for proteins related to cellular innate immunity and viral response, which are classified according to biological process, cellular component, and molecular function. (**C**) KEGG analysis of 42 differentially expressed proteins after screening for proteins related to cellular innate immunity and viral response, and the different signaling pathways involved in the regulation of these proteins, were classified.

**Figure 3 viruses-14-02587-f003:**
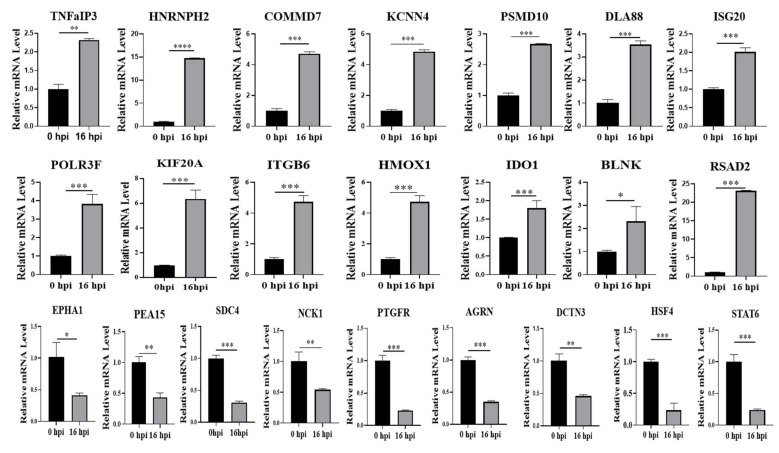
Validation of mRNA expression. A: Proteins were found to be differentially expressed between uninfected MDCK mock and infected MDCK H1N1 16 hpi cells based on proteomics screening. Of these proteins, 23 are involved in regulation of the host antiviral innate immune response. qPCR was used to validate these proteomics results at the mRNA level, using the 2^−ΔΔCt^ method and GAPDH as the internal reference gene. * Was used to determine statistically significant differences between groups; * *p* < 0.05, ** *p* < 0.01, *** *p* < 0.001, **** *p* < 0.0001.

**Figure 4 viruses-14-02587-f004:**
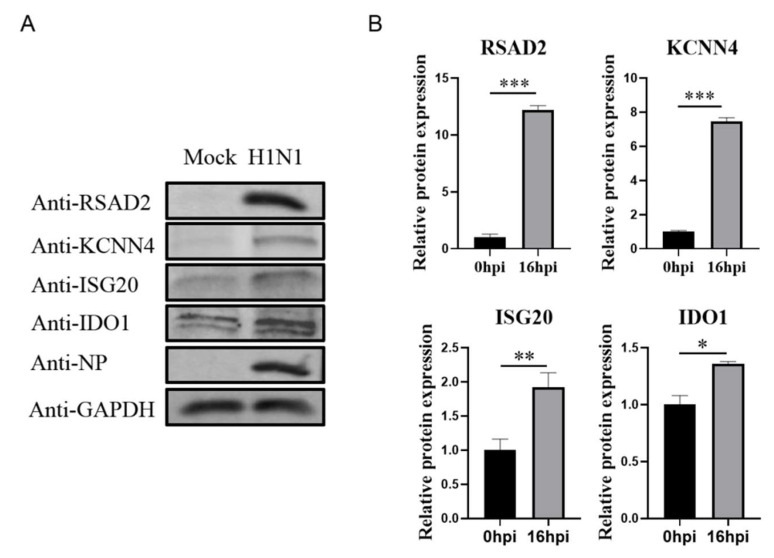
Verification of protein level expression of differential proteins. (**A**) The protein expression of RSAD2, IDO1, ISG20, and KCNN4 in infected MDCK H1N1 16 hpi cells was analyzed using western blot with GAPDH as the loading control. (**B**) Image software used for the gray value analysis of immunoblotting. * Was used to determine statistically significant differences between groups; * *p* < 0.05, ** *p* < 0.01, *** *p* < 0.001.

**Figure 5 viruses-14-02587-f005:**
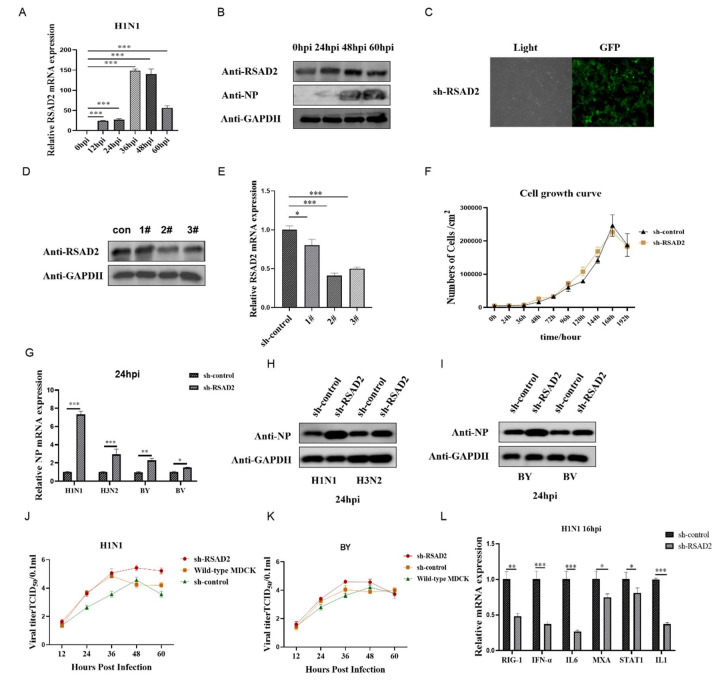
Knockdown of RSAD2 promotes influenza virus proliferation. (**A**) RSAD2 gene expression was detected in MDCK cells infected with IAV H1N1 at 0, 12, 24, 36, 48, and 60 h post infection (hpi) using qPCR. (**B**) RSAD2 protein expression was detected in MDCK cells infected with IAV H1N1 at 0, 24, 48, and 60 hpi using western blot. (**C**) Representative images of sh-RSAD2 with a GFP tag using a 40× fluorescence microscope are shown. (**D**) Protein expression of RSAD2 in sh-RSAD2 MDCK cells was measured using western blot to verify successful knockdown. (**E**) Gene expression of RSAD2 in sh-RSAD2 (#1, #2, #3) MDCK cells was measured using qPCR to verify successful knockdown. (**F**) The proliferation of sh-control and sh-RSAD2 cells was detected using the digestion technique. (**G**) Influenza A/Puerto Rico/8/34 (A/PR/8/34) H1N1 virus, A/Texas/50/2012 (H3N2) NYMCX-223A, B/Colorado/06/2017-like virus B (Victoria lineage), and B/Phuket/3073/2013-like virus B (Yamagata/16/88 lineage) NP gene expression were measured in sh-RSAD2 MDCK cells using qPCR. (**H,I**) Influenza A/Puerto Rico/8/34 (A/PR/8/34) H1N1 virus, A/Texas/50/2012 (H3N2) NYMCX-223A, B/Colorado/06/2017-like virus B (Victoria lineage), and B/Phuket/3073/2013-like virus B (Yamagata/16/88 lineage) NP protein expression were measured in sh-RSAD2 MDCK cells using western blot. (**J**,**K**) Growth kinetics of the influenza A/Puerto Rico/8/34 (A/PR/8/34) H1N1 virus, A/Texas/50/2012 (H3N2) NYMCX-223A, B/Colorado/06/2017-like virus B (Victoria lineage), and B/Phuket/3073/2013-like virus B (Yamagata/16/88 lineage) vaccine strain by TCID_50_ method. (**L**) Antiviral factor gene expression was measured in sh-RSAD2 MDCK cells using qPCR. GAPDH was used as the reference gene for qPCR and GAPDH was used as the loading control for western blot. * Was used to determine statistically significant differences between groups; * *p* < 0.05, ** *p* < 0.01, *** *p* < 0.001.

**Table 1 viruses-14-02587-t001:** Expression of significantly changed immune-response-related proteins in MDCK cells after influenza virus H1N1 infection.

Class	Gene ID	Protein ID	Gene Name	log_2_FC ^a^	*p* Value ^b^
UP	488368	A0A5F4BV48	Integrin alpha-V: beta-6 (ITGB6)	1.19293406	0.0000282
	486814	A0A5F4BZS9	B cell linker (BLNK)	1.45167078	0.0169468
	475574	A0A5F4D9Z0	Indoleamine 2, 3-dioxygenase 1 (IDO1)	3.15270072	0.0001613
	484006	A0A8C0M1H5	Tumor necrosis factor-alpha-inducing protein 3 (TNFAIP3)	1.82043059	0.0279786
	119868481	A0A8C0RFV7	Heterogeneous nuclear ribonucleoprotein H2 (HNRNPH2)	1.10931159	0.0220326
	612903	A0A5F4CVE0	COMM domain protein 7 (COMMD7)	1.04388446	0.0090970
	484464	A0A5F4DCH1	Potassium Calcium-Activated Channel Subfamily N Member 4 (KCNN4)	2.75287993	0.0001107
	474836	A0A6S4QBX3	MHC class I DLA-88 (DLA88)	1.47508145	0.0042902
	488729	A0A8C0M265	Interferon stimulated exonuclease gene 20 (ISG 20)	1.09012632	0.0033698
	442987	A0A8C0MMC9	Heme oxygenase 1 (HMOS1)	3.23512999	6.70E-05
	609005	A0A8C0NII9	Radical S-adenosyl methionine domain containing 2 (RSAD2)	2.55784784	0.0178478
	481014	A0A8C0NRZ7	Proteasome 26S subunit, non-ATPase 10 (PSMD10)	1.12860962	0.0266508
	477143	A0A8C0RSR8	Polymerase III subunit F (POLR3F)	1.99118698	0.0478663
	474693	A0A8C0MET1	Kinesin family member 20A (KIF20A)	3.05125558	1.62E-06
Down	482735	A0A8C0N3E9	EPH receptor A1 (EPHA1)	−3.22527176	0.0006628
	610113	A0A8C0NJA5	Proliferation and apoptosis adaptor protein 15 (PEA15)	−1.09485499	0.0463038
	485893	A0A8C0TU12	Syndecan 4(SDC4)	−2.32177021	0.0448164
	485677	A0A8C0TTD9	NCK adaptor protein 1(NCK1)	−1.09055648	0.0160281
	475809	A0A5F4CVB7	Prostaglandin F2 receptor inhibitor (PTGFRN)	−2.22617135	4.26E-05
	474750	A0A5F4C381	Dynactin subunit 3 (DCTN3)	−1.02874396	0.0105753
	489766	A0A8C0Z157	Heat shock transcription factor 4 (HSF4)	−1.61719123	0.0002577
	479574	A0A5F4C2A2	Agrin (AGRN)	−2.18378056	8.70E-08
	100142679	F1PAY3	Signal transducer and activator of transcription 6 (STAT6)	−1.00699288	0.0029116

^a^ The ratio of the expression levels between the two groups of samples, taking the logarithm to the base of 2. ^b^ The proteins that had statistically significant differences (*p* < 0.05).

## Data Availability

The mass spectroscopic proteomic data are shown in Appendix A.

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
