# Peer review of "RSAD2 Is an Effective Target for High-Yield Vaccine Production in MDCK Cells"

_viruses, 2022, doi:10.3390/v14112587_

Round 1
Reviewer 1 Report
the study presented by Qiao et al. gave a possible target for engineering cell lines producing high-yield viruses. The study was well designed but several key points need to be addressed:
major:
1. mock-infected MDCK cells harvested at 16hpi will be better in regarding differentially expressed proteins analysis.
2. in figure 5, authors utilized NA mRNA and protein levels as an indicator of viral replication. However, IAV growth curves in original MDCK, sh-control, and sh-RSAD2 cell lines are needed and should be involved in the figure to demonstrate viral replication and impacts of RSAD2.
other comments:
1. language should be polished by native speakers to make it clear and no grammar mistakes.
2. in the result section, not only make summarizing sentences but clearly describe and present the observation and results.
Author Response
November,9,2022
Dear Reviewer,
We would like to thank you for carefully review of and very helpful comments for our manuscript “RSAD2 is an effective target for high-yield vaccine production in MDCK cells” (Manuscript ID:2003533). These comments are very useful for improving our manuscript and also important for our future research works. We have studied the comments carefully. In response to your comments, we have corrected mistakes in revised manuscript. We hope this time our revised manuscript can meet the styles which are required by VIRUSES.
Thank you for your consideration of this revised manuscript for publication in
VIRUSES.
The point-by-point responses to reviewers’ comments are given as following.
Best regards.
Sincerely,
Dr. Xiaoming Yang,
Responses to reviewers’ comments
Responses to Referee:
Comments 1: Mock-infected MDCK cells harvested at 16hpi will be better in regarding differentially expressed proteins analysis.
Response: Thank you for the comments. We are very sorry for the ambiguity caused by our wrong labeling. The blank group is the mock group, and the 16hpi is the cell sample infected by the virus for 16 hours, which is the cell sample collected at the same time. We have made changes in the article. See Fig. 4 on page 10 of the manuscript.
Comments 2: In figure 5, authors utilized NA mRNA and protein levels as an indicator of viral replication. However, IAV growth curves in original MDCK, sh-control, and sh-RSAD2 cell lines are needed and should be involved in the figure to demonstrate viral replication and impacts of RSAD2.
Response: Thank you for the comments. TCID50 is an important method to evaluate influenza virus proliferation, according to your suggestion, we added the missing content in the article. See Fig. 5 on page 11 of the manuscript.
Comments 3: Language should be polished by native speakers to make it clear and no grammar mistakes. In the result section, not only make summarizing sentences but clearly describe and present the observation and results.
Response: Thank you for your comments. The manuscript has been polished and modified by the article polishing agency and according to your suggestion, we have made appropriate changes to the result section. For details, see the fields marked in red in the manuscript for details.

Reviewer 2 Report
The authors present a detailed proteomic analysis of adherent MDCK cells which is supported by limited transcriptomic analysis and knockdown experiments. Overall these results are of value to the field. However, as the authors point out the current use of MDCK cells to produce influenza antigen utilises adapted cell suspension cell lines in supplemented media. In the experiments presented the authors use adherent cell lines with limited description of the state of these cells with respect to logarithmic growth. Currently trivalent and quadrivalent influenza vaccines are use of seasonal vaccination to prevent influenza and monovalent pandemic infection. The authors analyse a single infection with A/Peurto Rico/8/34. This particular virus is well known to grow at very high levels and thus the inclusion of a single subtype/ single virus limits this manuscript. The manuscript would be much improved to analyse examples of H3N2, H5N1 and B viruses. Further demonstration of increased levels of NP does not necessarily reflect the level of antigen that may be generated. The Western blot in the final figure (G) could be anlaysed by densitometry.
The authors would be assisted with a review of English language with numerous examples of poor language use. For example:
Line 43 should be referenced
Line 45 cell suspension domestication – what does this mean? Poor use of language
Line 57 viral titre of IAV by tens of times - Poor use of language
Line 167/175/184 MDCK cells in good growth condition = logarithmic growth?
Line 190 the cell growth curve
Line 199 We used mass spectrometry analysis the differential protein expression poor language
Author Response
November,9,2022
Dear Reviewer,
We would like to thank you for carefully review of and very helpful comments for our manuscript “RSAD2 is an effective target for high-yield vaccine pro-duction in MDCK cells” (Manuscript ID:2003533). These comments are very useful for improving our manuscript and also important for our future research works. We have studied the comments carefully. In response to your comments, we have corrected mistakes in revised manuscript. We hope this time our revised manuscript can meet the styles which are required by VIRUSES.
Thank you for your consideration of this revised manuscript for publication in
VIRUSES.
The point-by-point responses to reviewers’ comments are given as following.
Best regards.
Sincerely,
Dr. Xiaoming Yang,
Responses to reviewers’ comments
Responses to Referee:
Comments 1: The authors present a detailed proteomic analysis of adherent MDCK cells which is supported by limited transcriptomic analysis and knockdown experiments. Overall these results are of value to the field. However, as the authors point out the current use of MDCK cells to produce influenza antigen utilises adapted cell suspension cell lines in supplemented media. In the experiments presented the authors use adherent cell lines with limited description of the state of these cells with respect to logarithmic growth. Currently trivalent and quadrivalent influenza vaccines are use of seasonal vaccination to prevent influenza and monovalent pandemic infection. The authors analyse a single infection with A/Peurto Rico/8/34. This particular virus is well known to grow at very high levels and thus the inclusion of a single subtype/ single virus limits this manuscript. The manuscript would be much improved to analyse examples of H3N2, H5N1 and B viruses. Further demonstration of increased levels of NP does not necessarily reflect the level of antigen that may be generated. The Western blot in the final figure (G) could be anlaysed by densitometry.
Response: Thank you for your comments. According to your suggestion, we added the results of other influenza viruses in the manuscript, including the effect of knockdown of RSAD2 on the replication and proliferation of Influenza A/Puerto Rico/8/34 (A/PR/8/34) H1N1 virus、 A/Texas/50/2012(H3N2) NYMCX-223A、B/Colorado/06/2017-like virus B (Victoria lineage) and B/Phuket/3073/ 2013-like virus B (Yamagata/16/88 lineage).TCID50 is an important method to evaluate influenza virus proliferation, we added the missing content in the article. See Fig. 5 on page 11 of the manuscript.
Comments 2: The authors would be assisted with a review of English language with numerous examples of poor language use. For example: Line 43 should be referenced; Line 45 cell suspension domestication what does this mean? Poor use of language; Line 57 viral titre of IAV by tens of times - Poor use of language; Line 167/175/184 MDCK cells in good growth condition = logarithmic growth? Line 190 the cell growth curve; Line 199 We used mass spectrometry analysis the differential protein expression poor language.
Response: Thank you for the comments. According to your suggestion, we check and modify the content in the manuscript. For details, see the fields marked in red in the manuscript for details.

Round 2
Reviewer 1 Report
Thank you and the revised MS addressed my concerns.
Reviewer 2 Report
Thankyou for modifying the manuscript. I believe this is much improved with respect to a mor general influence on influenza subtypes